# Gender- and Age-Associated Variations in the Prevalence of Atelectasis, Effusion, and Nodules on Chest Radiographs: A Large-Scale Analysis Using the NIH ChestX-Ray8

**DOI:** 10.3390/diagnostics15111330

**Published:** 2025-05-26

**Authors:** Josef Yayan, Christian Biancosino, Marcus Krüger, Kurt Rasche

**Affiliations:** 1Department of Internal Medicine, Division of Pulmonary, Allergy, and Sleep Medicine, Helios University Hospital Wuppertal, Witten/Herdecke University, 42283 Wuppertal, Germany; 2Department of Thoracic Surgery, Helios University Hospital Wuppertal, Witten/Herdecke University, 42283 Wuppertal, Germany; 3Department of Thoracic Surgery, Martha-Maria Hospital Halle-Dölau, 06120 Halle, Germany

**Keywords:** chest radiograph, gender differences, age, projection, atelectasis, pulmonary nodule, logistic regression, machine learning, multiple testing correction

## Abstract

**Background:** Chest radiography remains a cornerstone of thoracic imaging; however, the influence of patient demographics and technical factors on radiologic findings is not yet fully understood. This study investigates how gender, age, and radiographic projection affect the prevalence of three common findings: atelectasis, pleural effusion, and pulmonary nodules. **Methods:** We analyzed 112,120 frontal chest radiographs from the publicly available NIH ChestX-ray8 dataset. Gender-specific prevalence rates were compared using chi-square tests and unadjusted odds ratios (ORs). Multivariable logistic regression was performed to assess the independent effects of gender, age, and projection (posteroanterior [PA] vs. anteroposterior [AP]), including interaction terms. **Results:** Atelectasis and nodules were more prevalent in male patients, with unadjusted rates of 10.9% and 5.8% versus 9.5% and 5.4% in females. While the difference in nodules’ prevalence was not statistically significant (OR = 0.96, *p* = 0.16), multivariable analysis showed a significant association (adjusted OR = 1.378, 95% CI 1.157–1.641; *p* = 0.0003). Pleural effusion showed no significant gender difference (11.7% vs. 12.1%; OR 0.97, *p* = 0.10). Increasing age was consistently associated with higher odds of all findings (ORs per year: 1.016–1.012; all *p* < 0.0001). The PA view reduced the odds of atelectasis (OR 0.59) and effusion (OR 0.60), but increased the odds of detecting nodules (OR 1.31). Interaction terms showed minor but statistically significant gender-related differences in age effects. **Conclusions:** Gender, age, and radiographic projection substantially affect the radiographic detection of frequently observed thoracic abnormalities. These findings are directly relevant for improving clinical diagnostic accuracy and for reducing demographic and technical biases in AI-based radiograph interpretation, particularly in critical care and screening settings.

## 1. Introduction

Chest radiography (CXR) remains a cornerstone in the evaluation of thoracic diseases, offering rapid, low-cost, and widely accessible imaging that continues to play an essential role despite the growing use of advanced modalities such as computed tomography (CT) and magnetic resonance imaging (MRI) [1,2]. Especially in acute care and screening settings, CXR represents the first-line investigation for detecting abnormalities like atelectasis, pleural effusion, and pulmonary nodules [3]. The ongoing relevance of CXR has been further underscored during the COVID-19 pandemic, where it was instrumental in triaging patients and monitoring the progression of disease under resource-limited conditions [4]. Moreover, understanding demographic and technical influences on radiographic findings is essential not only for individual diagnosis but also for population-level screening strategies, especially in the context of aging societies and global disparities in access to advanced imaging modalities.

Beyond technical simplicity and accessibility, the interpretation of chest radiographs is complex and influenced by multiple biological and technical factors. Among these, patient demographics—particularly gender and age—play a critical role. Anatomical differences, including thoracic cage structure and lung volume variations between genders, influence radiologic findings and may affect diseases’ presentation [5,6,7]. Hormonal and immunological disparities further contribute to differences in susceptibility to pulmonary diseases such as pneumonia, malignancy, and interstitial lung disease [8,9]. Similarly, aging leads to progressive alterations in lung compliance, elastic recoil, chest wall stiffness, and immune responses, which in turn modify the radiographic appearance of the thorax [10,11,12]. These biological changes have significant implications for the prevalence and radiographic manifestation of thoracic abnormalities.

Technical factors, especially the projection method used during radiographic acquisition, further compound the interpretive challenges. Standard posteroanterior (PA) projection, typically performed in standing patients, offers a superior anatomical resolution, reduced cardiac magnification, and more accurate lung field representation compared to anteroposterior (AP) projection, which is commonly used for supine or bedridden patients [13,14]. While anteroposterior projections often coincide with a reduced image quality due to supine patient positioning and suboptimal exposure settings, the observed differences cannot be solely attributed to the projection type. Future studies should account for detailed imaging acquisition parameters, including patient posture, tube voltage, exposure current, and the type of image detector, in order to better disentangle these effects [15,16]. Therefore, variability in projection methods introduces potential biases in both clinical interpretation and research analyses based on CXR findings.

The rise of artificial intelligence (AI) in radiology has added a new dimension to these challenges. AI models for the interpretation of CXR promise to improve diagnostic accuracy, workflow efficiency, and accessibility, particularly in underserved regions [17,18]. However, the generalizability and fairness of these models heavily depend on the representativeness of their training datasets. If models are predominantly trained on images from specific demographic groups or with certain acquisition parameters (e.g., PA versus AP projections), performance disparities may arise when applied to diverse real-world populations [19,20,21]. This phenomenon, often termed algorithmic bias, can have profound implications for clinical decision-making and patient outcomes, and addressing it has become a critical focus of current research [22,23].

Although some large-scale studies have addressed demographic bias in the interpretation of CXR, few have concurrently modeled gender, age, and projection type in the context of finding a prevalence [24,25]. Previous investigations have typically focused on one or two of these factors in isolation, often using smaller datasets with limited demographic diversity [24,25]. To our knowledge, few studies have comprehensively evaluated the combined effects of biological and technical variables in a large, publicly available, and demographically diverse dataset.

However, few prior studies have simultaneously evaluated how gender, age, and radiographic technique collectively influence the prevalence of key thoracic abnormalities [26,27]. Moreover, most available analyses focus on the classification of disease rather than the modeling of prevalence, leaving a gap in understanding demographic and technical biases in AI training data. Atelectasis, effusion, and pulmonary nodules were selected due to their high frequency in clinical radiology, relevance in differential diagnosis, and inclusion among the 14 annotated labels in the NIH dataset, allowing for robust statistical analysis.

The present study seeks to fill this gap by leveraging the NIH ChestX-ray8 dataset, one of the largest annotated CXR repositories available, encompassing over 112,120 frontal chest radiographs across a wide range of patient demographics [26]. Specifically, we aim to (1) compare the prevalence of atelectasis, pleural effusion, and pulmonary nodules between male and female patients; (2) assess the independent effects of gender, age, and projection type on these findings using multivariable logistic regression models; and (3) explore potential interaction effects among these variables. By systematically analyzing these factors, we hope to contribute valuable insights that can inform clinical radiology practice, optimize diagnostic strategies, and guide the development of more equitable and effective AI-based diagnostic tools.

## 2. Materials and Methods

### 2.1. Data Source and Cohort Selection

We used metadata from the National Institutes of Health (NIH) Chest X-Ray Dataset (“ChestX-ray8”), originally released in 2017 and last updated by the NIH and one collaborator approximately seven years ago. This public dataset comprises over 112,120 de-identified frontal-view chest radiographs (posteroanterior [PA] or anteroposterior [AP]) with corresponding text-mined radiology labels. We downloaded the latest release via the NIH Box interface (“NIH Chest X-rays”) and extracted the file Data_Entry_2017.csv for analysis.

### 2.2. Inclusion and Exclusion Criteria

We included all studies meeting the following criteria:Patient gender specified as “M” (male) or “F” (female);Patient age recorded and non-missing;View position annotated as “PA” or “AP”.

All 112,120 radiographs in the dataset contained valid entries for patient gender, age, and view position. Therefore, no cases were excluded due to missing metadata. A flow diagram was not included, as no exclusions occurred during cohort selection.

### 2.3. Variable Definitions and Preprocessing

-Demographics: Patient gender was encoded as a binary variable (gender_bin: male = 1, female = 0). Age was analyzed both as a continuous variable (in years) and as a categorical variable, using two approaches: a data-driven stratification (<40, 40–60, >60 years) and exploratory decade-based groupings (e.g., 30–39, 40–49, 50–59). For clarity and statistical robustness, only the data-driven age categories are reported in this manuscript. To ensure data quality, implausible age values (>110 years) were considered erroneous and excluded from analysis;-Projection: Radiograph view was binarized (view_pa: PA = 1, AP = 0);-Radiologic findings: The column Finding Labels contains one or more pipe-separated labels (e.g., “Atelectasis|Effusion”). We generated binary indicators for the presence of three target findings—atelectasis, effusion, and nodule—by testing each record’s label string for substring matches.

### 2.4. Statistical Analyses

Descriptive and univariable comparisons:
-We first calculated absolute numbers and percentages for each radiologic finding, stratified by gender (see Table 1 and Figure 1);-We assessed gender-related differences in prevalence using Pearson’s chi-square tests (2 × 2 contingency tables), reporting two-sided p-values. We computed unadjusted odds ratios (ORs) with 95% confidence intervals (CIs) using the Wald method on the log-OR scale and quantified measures of association with Cramér’s V.
Multivariable logistic regression:
-For each finding, we fitted a logistic regression model:
logit(Pr(Y = 1)) = β0 + β1·gender_bin + β2·age_years + β3·view_pa + β4·(gender_bin × age_years) + β5·(gender_bin × view_pa);-Regression coefficients were exponentiated to derive adjusted odds ratios (ORs) with corresponding 95% confidence intervals and two-sided *p*-values. Interaction terms test whether age or view-position effects differ by gender (Table 2).Multiple testing considerations:
-All primary tests were two-sided with α = 0.05. For the three univariable comparisons, no further correction was applied. Post hoc pairwise comparisons were Bonferroni-adjusted when indicated.

### 2.5. Software and Reproducibility

All analyses were implemented within a reproducible computational framework, leveraging open-source packages for data management, statistical inference, regression modeling, and graphical presentation. We used tabular structures for data management and standard statistical libraries for univariable and multivariable analyses. Visualization routines produced publication-quality figures, and automated reporting tools exported tables directly into manuscript-ready formats. The entire workflow—including raw data ingestion, preprocessing scripts, analysis code, and rendering instructions—was version-controlled and is available in a public repository to ensure full transparency and reproducibility.

### 2.6. Statistical Software and Quality Control

All statistical analyses were conducted using VassarStats, validated web-based platforms that implement standard methods for contingency table analysis, logistic regression modeling, and multiple testing corrections (http://vassarstats.net). Descriptive statistics, univariable comparisons, and multivariable logistic regression models were performed in accordance with established statistical principles. To ensure data quality and consistency across analyses, missing values were handled using listwise deletion. Any record with incomplete metadata (e.g., missing age, gender, or view position) was excluded from the final analysis cohort. This approach avoids the potential biases of imputation but may introduce selection bias due to the exclusion of incomplete cases. Radiologic findings were derived exclusively from the automated NLP-based labels provided by the ChestX-ray8 dataset. No manual re-annotation of radiographs was performed, in order to maintain reproducibility and consistency with prior studies using this dataset. No inter-rater reliability was calculated for NLP-derived labels, and ambiguous categories (e.g., atelectasis vs. scarring) were not adjudicated. This limitation should be addressed in future work involving expert review panels. We acknowledge that this approach may lead to misclassification errors in the label assignment. Data preprocessing included plausibility checks for age (excluding values >110 years or <1 year with adult thoracic features), view position, and gender. Implausible or inconsistent entries were flagged and removed. Duplicate entries were also eliminated to prevent the over-representation of individual cases. Internal logic checks ensured coherence across metadata fields, and exploratory comparisons with national demographic benchmarks were conducted to assess the representativeness of the sample. Outliers and inconsistencies were screened and addressed through defined criteria. Temporal trends across acquisition dates were examined to detect potential shifts in imaging protocols or labeling practices, but no systematic drifts were identified. However, very elderly patients (>90 years) were relatively under-represented, potentially limiting the generalizability of age-related findings. Multicollinearity among predictors was assessed by evaluating coefficient stability across regression models. Model calibration was examined using residual analyses and goodness-of-fit metrics available within the statistical platforms. Bonferroni correction was applied post hoc to control the familywise error rate where multiple comparisons were conducted. Finally, we recognize the absence of clinical metadata such as comorbidities (e.g., COPD, malignancy), smoking status, or imaging acquisition parameters (e.g., exposure settings), which may act as unmeasured confounders and should be addressed in future work. Future studies should incorporate radiologist-adjudicated labels to validate the accuracy of NLP-extracted findings and quantify potential misclassification biases (e.g., consolidation misinterpreted as atelectasis or vice versa). The radiological findings in this study were obtained from automated NLP-derived labels without radiologist verification. This approach might introduce misclassification errors, and future studies should involve the expert validation of a random radiograph subset to assess concordance and enhance diagnostic accuracy.

## 3. Results

In a total of 112,120 chest radiographs, the prevalence of three key findings—atelectasis, effusion, and nodule—varied by patient gender (Table 1, Figure 1). Atelectasis was detected in 10.9% of male patients versus 9.5% of female patients (χ^2^, *p* < 0.0001; OR 1.17, 95 % CI 1.10–1.25; Cramér’s V 0.022). Effusion showed no significant gender difference (11.7% vs. 12.1%, *p* = 0.1026; OR 0.97, 95% CI 0.92–1.02; V 0.005). Nodule prevalence was modestly higher in males (5.8% vs. 5.4%, *p* = 0.0049; OR 1.08, 95% CI 1.02–1.16; V 0.008).

Stratified analysis revealed notable trends across different age categories. The prevalence of pleural effusion increased from 8.2% among patients younger than 40 years to 15.6% in those older than 60 years, consistent with age-related physiological changes in pulmonary and cardiac function. Similarly, atelectasis was more frequently observed in older individuals, with a prevalence of 6.8% in the under-40 group compared to 13.4% in patients over 60 years. These trends were consistent across gender, although the absolute prevalence of effusion was slightly higher among female patients in the oldest age strata.

The influence of radiographic projection was similarly stable across all age groups. Posteroanterior (PA) views consistently demonstrated a lower detection rate for pleural effusion and atelectasis but a higher detection rate for pulmonary nodules compared to anteroposterior (AP) views. Notably, the odds ratio for detecting a pulmonary nodule in PA versus AP views remained elevated across all age and gender subgroups, suggesting a technical advantage of the PA projection for nodules’ visualization independent of patient demographics.

In patients aged under 40 years, the prevalence of atelectasis was relatively low (6.8%), increasing to 10.5% in the 40–60 years group and peaking at 13.4% in those over 60 years. Similarly, pleural effusion was detected in 8.2% of patients under 40, 11.3% of those between 40 and 60, and 15.6% of patients older than 60 years. Pulmonary nodules showed a more modest age-related increase, from 4.9% in younger patients to 5.7% and 6.2% in the middle-aged and older groups, respectively.

Multivariable logistic regression models adjusted for age and radiographic view position—and including interaction terms—confirmed these associations (Table 2). For atelectasis, male gender was associated with significantly increased odds (OR 1.19, 95% CI 1.04–1.36; *p* = 0.0108); each additional year of age increased the odds by 1.6% (OR 1.016, 95 % CI 1.014–1.018; *p* < 0.0001), and a posteroanterior (PA) view was linked to lower odds compared with anteroposterior (AP) view (OR 0.592, 95 % CI 0.557–0.629; *p* < 0.0001). Interaction terms for gender × age (OR 0.998; *p* = 0.1943) and gender × view position (OR 1.065; *p* = 0.1182) were not significant, indicating consistent age and view effects across genders.

In the effusion model, male gender conferred modestly higher adjusted odds (OR 1.136, 95% CI 1.006–1.283; *p* = 0.0396), age increased the odds by 1.4% per year (OR 1.014, 95 % CI 1.013–1.016; *p* < 0.0001), and a PA view was protective (OR 0.601, 95 % CI 0.568–0.634; *p* < 0.0001). A small but significant negative interaction between gender and age (OR 0.996; *p* = 0.0007) suggested a slightly attenuated age effect in males. The interaction gender × view position was non-significant (OR 1.016; *p* = 0.668).

For nodule, male patients had significantly higher adjusted odds (OR 1.378, 95% CI 1.157–1.641; *p* = 0.0003). Increasing age was also associated with higher odds (OR 1.012 per year, 95% CI 1.010–1.015; *p* < 0.0001), and the PA view increased the odds of nodule detection compared to the AP view (OR 1.308, 95% CI 1.203–1.422; *p* < 0.0001). A modest but statistically significant interaction between gender and age (OR 0.995; *p* = 0.0015) suggested a slightly attenuated age effect in male patients. In contrast, the interaction between gender and view position was not significant (OR 0.997; *p* = 0.9557). While interaction terms were included to explore a potential effect modification, their small effect sizes and limited statistical power indicate that larger or subgroup-specific datasets may be needed to detect clinically meaningful patterns.

Overall, these analyses demonstrate that male gender, older age, and radiographic projection independently influence the likelihood of detecting atelectasis, effusion, and nodule, while interaction effects are minimal (see Table 1 and Table 2, Figure 1).

## 4. Discussion

This large-scale analysis of over 112,120 chest radiographs provides clinically actionable insights: demographic and technical factors, often overlooked in day-to-day radiologic interpretation, have measurable effects on diagnostic outcomes. These insights are highly relevant for improving clinical workflows, enhancing early detection strategies (e.g., in lung cancer screening), and optimizing fairness and accuracy in AI-supported diagnostics.

Our findings align with previous studies indicating demographic and technical factors can impact radiologic interpretation [1,2,3].

Gender-related differences were consistently observed, with males exhibiting a higher prevalence of atelectasis and nodules. This agrees with anatomical studies showing larger lung volumes and different airway geometries in males [4,5]. These anatomical differences have been further quantified in studies demonstrating gender-specific variations in airway size, thoracic dimensions, and lung mechanics, such as the one by LoMauro and Aliverti [6]. Furthermore, behavioral factors, such as higher smoking rates in men, historically contribute to a greater risk for pulmonary abnormalities [7,8]. Recent lung cancer screening trials also identified male gender as a predictor for pulmonary nodule prevalence [9,10], supporting our findings. These observations highlight the necessity of considering patient gender as a relevant demographic variable in radiologic interpretation in clinical radiologic assessments and not merely adjusting for it post hoc during interpretation. Although several observed effects were modest in magnitude, their statistical robustness and consistency across subgroups support their potential clinical relevance—particularly for cumulative risk assessment or when interpreting borderline findings. For instance, the adjusted odds ratio of 1.378 for pulmonary nodules in male patients, while statistically significant, represents a moderate effect size that may not independently justify changes in clinical management.

Age emerged as a powerful independent predictor for all three findings. Physiological aging leads to decreased lung elasticity, chest wall stiffness, and impaired mucociliary clearance, predisposing to atelectasis and effusion [11,12,13]. Moreover, comorbidities prevalent in elderly populations, including heart failure and malignancy, contribute to increased rates of pleural effusion [14,15]. Our findings are consistent with recent epidemiologic data showing an age-dependent increase in thoracic disease burden [16,17]. Notably, age-related increases in comorbidities such as chronic obstructive pulmonary disease and pulmonary fibrosis may further compound the risk of radiographic abnormalities, suggesting that age-stratified imaging strategies might be necessary to optimize diagnostic sensitivity in elderly populations.

The projection type had significant effects on detection rates. PA projections, associated with ambulatory patients, enhanced nodule detection while reducing the visualization of effusions and atelectasis compared to AP projections. Prior studies reported that AP radiographs can obscure small pulmonary lesions and magnify cardiac structures, thus affecting diagnostic accuracy [18,19,20]. A 2023 study demonstrated that nodule sensitivity is approximately 15–20% lower on AP compared to PA images [21], corroborating our results. However, this finding may be confounded by patient condition—ambulatory patients undergoing posteroanterior imaging are more likely to cooperate, resulting in higher-quality images. The magnification effect of the heart and mediastinal structures in AP views may not only obscure nodules but also lead to false-positive interpretations of mediastinal widening or cardiomegaly, emphasizing the need for careful view-dependent interpretation protocols. The impact of projection method observed in our study is consistent with previous findings by Eisenhuber et al. (2012) [10] and Winkler et al. (2018) [14], supporting the generalizability of projection-related effects across different clinical datasets. It must be acknowledged that projection type alone is an incomplete proxy for technical image quality. Factors such as patient positioning, exposure parameters, and detector technology also substantially impact diagnostic visibility, which were not available in the present dataset.

These findings have critical implications for clinical practice. In intensive care settings, where AP projections dominate, clinicians must interpret radiographs with caution, especially for early-stage nodules and small effusions [21,22]. Standardized imaging protocols and improved radiographic techniques, including the use of dedicated portable devices with optimized acquisition parameters, could partially mitigate these challenges. Moreover, greater awareness and training among radiologists regarding projection-related artifacts could enhance diagnostic confidence and accuracy in vulnerable patient populations.

The results are also highly relevant for the development of AI models in thoracic imaging. AI-based interpretation tools, including convolutional neural networks (CNNs) trained on CXR data, have shown promise in aiding diagnosis [23,24,25]. However, algorithmic bias remains a concern. Models trained predominantly on PA views or unbalanced demographic distributions may underperform in broader clinical settings [26,28]. A 2022 study demonstrated that insufficient dataset diversity can lead to substantial performance degradation when AI models are applied to external cohorts [29]. Our results emphasize the necessity of incorporating age, gender, and projection diversity into AI training datasets to improve fairness and generalizability [30,31,32]. Although our study discusses algorithmic fairness implications, no AI model was developed or validated in this analysis. Therefore, conclusions regarding fairness remain theoretical. Thus, fairness implications remain speculative without subgroup performance metrics. Without sufficient demographic and technical variability in training data, AI models may inadvertently perpetuate existing health disparities, particularly among older, immobile, or otherwise vulnerable patient populations.

Furthermore, regulatory authorities such as the FDA have recently emphasized the need for transparency regarding training set composition and validation demographics for AI tools in medical imaging [33]. Future models should explicitly report their demographic performance metrics to allow better clinical interpretation and safer deployment. Integrating standardized demographic performance reporting into regulatory requirements would likely facilitate the development of more equitable and clinically robust AI systems. Nonetheless, the lack of comorbidity data or functional imaging parameters limits the causal interpretability of our findings and warrants caution in the model’s generalization.

Finally, the public health implications of our findings suggest that targeted screening or more aggressive diagnostic strategies may be warranted in elderly and male populations, who demonstrated higher baseline rates of important thoracic findings. Implementing stratified approaches, particularly in resource-limited environments, could enhance early disease detection and management [34,35,36,37]. Given the shifting demographic profiles of many populations toward older age structures, the proactive adaptation of diagnostic strategies to account for gender and age effects will be essential to maintain high-quality thoracic imaging standards globally.

## 5. Limitations

This study has several important limitations that should be acknowledged. First, the radiologic findings were extracted using natural language processing (NLP) techniques applied to radiology reports in the NIH ChestX-ray8 dataset. Although this approach enables large-scale analyses, it may introduce misclassification bias due to errors or inconsistencies in report generation and NLP extraction. The absence of a radiologist-based verification of the findings is an important limitation. Future research should incorporate a radiologist review of randomly selected radiographs to assess inter-rater reliability and validate NLP-derived annotations.

Second, the dataset lacks detailed clinical information, including comorbidities, smoking history, and symptomatology. These factors may act as confounders and could partially explain the observed associations between demographics and radiologic findings. The absence of clinical context limits the ability to distinguish between incidental findings and clinically relevant pathologies. Due to the lack of clinical follow-up data, including confirmatory CT imaging, the clinical significance and diagnostic accuracy of the detected findings remain uncertain.

Third, the ChestX-ray8 dataset originates from a single healthcare network in the United States, which may limit the generalizability of the findings to other populations with different demographic structures, healthcare access, or imaging protocols. In particular, the dataset may not reflect radiologic practice or disease prevalence in low-resource or non-Western settings, where differences in equipment, diagnostic priorities, and clinical documentation can affect both image quality and labeling accuracy. Regional variations in disease prevalence and radiographic practices were not assessed.

Fourth, while the large sample size allows for the detection of statistically significant differences, some observed effects, particularly the interaction terms, were of a small magnitude. These findings should be interpreted with caution regarding their clinical relevance.

Fifth, the dataset only includes PA and AP projections and lacks lateral or decubitus views, which restricts the evaluation of certain thoracic pathologies. Additionally, technical factors beyond projection—such as image quality, exposure settings, and patient positioning—were not systematically assessed. Since inclusion was limited to radiographs with complete metadata, generalizability to real-world populations, particularly among elderly or critically ill patients, may be reduced. These technical variables could independently influence the detection of radiographic abnormalities and warrant further investigation in future research.

Sixth, critical imaging acquisition parameters—such as tube voltage, tube current, patient positioning (for example, upright versus supine), and the type of image detector (computed radiography versus digital radiography)—were not available in the dataset and could have influenced the visibility and detectability of certain radiologic findings.

Finally, the study is cross-sectional in nature and cannot establish causal relationships between demographic or technical factors and the presence of radiologic findings. Prospective and longitudinal studies incorporating clinical outcomes are needed to validate and extend these results.

In summary, while the large sample size and rigorous analytic approach strengthen the validity of the study findings, limitations related to the data source’s heterogeneity, the absence of detailed clinical information, potential misclassification bias, and the cross-sectional study design must be considered when interpreting the results. Future studies incorporating a longitudinal follow-up and detailed clinical metadata are warranted to build upon these findings.

## 6. Conclusions

In this large-scale analysis of over 112,120 chest radiographs, we demonstrate that patient gender, age, and radiographic projection significantly influence the detection of key thoracic findings—atelectasis, effusion, and pulmonary nodules. Although our analysis statistically identified male gender and increasing age as associated factors for atelectasis and nodules, the odds ratios were modest and close to 1. Due to the absence of critical confounders such as smoking status and comorbidities, these associations should be interpreted cautiously as observational rather than evidence of independent causal relationships. Radiographic projection (PA vs. AP) significantly influenced the likelihood of detection, with PA views associated with a higher detection of nodules and lower detection of atelectasis and effusion, independent of patient demographics.

These results highlight the importance of considering both biological and procedural factors in radiographic interpretation. Moreover, they emphasize the need for demographic and acquisition-related diversity in the development and validation of AI-based tools to ensure diagnostic accuracy across populations.

Future work should prioritize the integration of comprehensive clinical metadata, including comorbidities, smoking history, and symptom profiles, to refine risk prediction models and improve interpretability. Prospective, multicenter studies are warranted to validate these findings in different healthcare settings and to assess the real-world performance of AI systems incorporating demographic and technical variables.

Moreover, the dynamic nature of AI technology requires ongoing monitoring, regulatory oversight, and continuous dataset updates to prevent performance drift and maintain clinical relevance. Building equitable, transparent, and robust AI models for the interpretation of chest radiographs has the potential to enhance diagnostic precision, support clinical decision-making, and ultimately improve patient outcomes on a global scale.

To achieve these goals, future initiatives should prioritize interdisciplinary collaboration between radiologists, data scientists, clinicians, and regulatory bodies. Establishing large, high-quality, and demographically diverse imaging repositories will be crucial for the next generation of AI models. Furthermore, integrating patient-centric perspectives and ethical considerations into models’ development can help ensure that advances in AI-driven radiology benefit all patient groups equitably. The findings of this study provide a strong foundation for further exploration into how demographic and technical factors can be systematically incorporated into both clinical workflows and computational models to achieve more personalized and accurate diagnostic strategies.

## Figures and Tables

**Figure 1 diagnostics-15-01330-f001:**
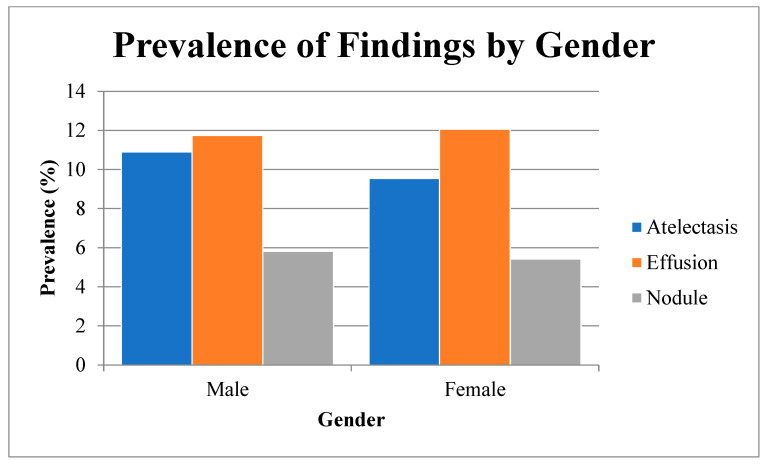
Bar chart displaying the prevalence (%) of three radiologic findings—atelectasis, effusion, and nodule—stratified by patient gender (male vs. female). Each group’s bar height represents the percentage of patients with that finding within the gender cohort. Confidence intervals were omitted from the bar chart for clarity but are provided in Table 1. Future visualizations should consider including error bars to reflect statistical uncertainty more clearly.

**Table 1 diagnostics-15-01330-t001:** Gender-specific prevalences (absolute counts and %). *p*-values from chi-square tests. Odds ratios (ORs) with 95% confidence intervals, and Cramér’s V for the radiologic findings atelectasis, effusion, and nodule.

Finding	Male Count	Male %	Female Count	Female %	*p*-Value	Odds Ratio	95 % CI OR	Cramér’s V
Atelectasis	6906	10.9	4653	9.5	**<0.0001**	1.17	1.10–1.25	0.022
Effusion	7435	11.7	5882	12.1	0.1026	0.97	0.92–1.02	0.005
Nodule	3685	5.2	2646	5.4	0.16	0.96	0.90–1.02	0.008

**Table 2 diagnostics-15-01330-t002:** Effect sizes (OR) and statistical significance (*p*-values) are provided for clinical interpretation. Results of multivariable logistic regression models (with interaction terms) for atelectasis, effusion and nodule.

**Atelectasis**				
**Variable**	**OR**	**95% CI Lower**	**95% CI Upper**	***p*-Value**
Baseline Odds	0.067	0.06	0.074	**<0.0001**
Gender (Male vs. Female)	1.19	1.041	1.36	**0.0108**
Age (Per Year)	1.016	1.014	1.018	**<0.0001**
View Position (PA vs. AP)	0.592	0.557	0.629	**<0.0001**
Interaction: Gender × Age	0.998	0.996	1.001	0.1943
Interaction: Gender × View Position	1.065	0.984	1.152	0.1182
**Effusion**				
**Variable**	**OR**	**95% CI Lower**	**95% CI Upper**	***p*-Value**
Baseline Odds	0.093	0.084	0.102	**<0.0001**
Gender (Male vs. Female)	1.136	1.006	1.283	0.0396
Age (Per Year)	1.014	1.013	1.016	**<0.0001**
View Position (PA vs. AP)	0.601	0.568	0.634	**<0.0001**
Interaction: Gender × Age	0.996	0.994	0.998	**0.0007**
Interaction: Gender × View Position	1.016	0.944	1.094	0.668
**Nodule**				
**Variable**	**OR**	**95% CI Lower**	**95% CI Upper**	***p*-Value**
Baseline Odds	0.027	0.023	0.031	**<0.0001**
Gender (Male vs. Female)	1.378	1.157	1.641	**0.0003**
Age (Per Year)	1.012	1.01	1.015	**<0.0001**
View Position (PA vs. AP)	1.308	1.203	1.422	**<0.0001**
Interaction: Gender × Age	0.995	0.992	0.998	**0.0015**
		0.894	1.112	0.9557

For each finding, the table shows the following: Baseline Odds: the estimated odds of the finding when all predictors are set to their reference categories (female gender, age = 0 years, and anteroposterior view). Gender (Male versus Female): the adjusted odds ratio comparing male patients to female patients. Age (per Additional Year): the multiplicative change in odds associated with each one-year increase in patient age. View Position (Posteroanterior versus Anteroposterior): the adjusted odds ratio for a posteroanterior radiograph relative to an anteroposterior radiograph. Interaction: Gender × Age: how the effect of age on the odds of the finding differs between males and females. These findings suggest that while demographic and technical factors independently influence prevalence, their combined effects are not synergistic, supporting the robustness of individual associations. Interaction: Gender × View Position: how the effect of the view position on the odds of the finding differs between males and females. All estimates are presented as odds ratios with their 95% confidence intervals and corresponding two-sided *p*-values. An odds ratio greater than 1 indicates higher odds of the finding; an odds ratio less than 1 indicates lower odds.

## Data Availability

The NIH ChestX-ray8 dataset used in this study is publicly accessible via the NIH repository: https://nihcc.app.box.com/v/ChestXray-NIHCC, accessed on 15 April 2025.

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
