# Peer review of "Gender- and Age-Associated Variations in the Prevalence of Atelectasis, Effusion, and Nodules on Chest Radiographs: A Large-Scale Analysis Using the NIH ChestX-Ray8"

_diagnostics, 2025, doi:10.3390/diagnostics15111330_

Round 1

Reviewer 1 Report

Comments and Suggestions for Authors

To the Authors,

I want to express my gratitude to the author for submitting this article, which highlights a crucial radiological finding and correlates it with gender and age.

The authors must address and respond to the following inquiries and requests.

  • Title:
  • Using Gender instead of Sex would be more appropriate.
  • Referring to the dataset studied (NIH Chestx-ray8) would be more comprehensive.
  • Keywords:

Expand keywords to include methodology such as “logistic regression,” “machine learning,” and statistical considerations such as “multiple testing correction.”

  • In the introduction section:
  • Please more clearly articulate the gap in current literature that this study addresses.
  • Provide a concise rationale for focusing on the selected findings: Atelectasis, Effusion, and Nodule.
  • Reference relevant prior studies for direct comparison.
  • Methodology:
  • Please justify the use of listwise deletion to handle missing data and discuss its potential impact on selection bias.
  • Clearly state that no manual observations were conducted and address any potential bias in the misclassification of automated labels.
  • Provide more detail on how outliers and data inconsistencies were identified and handled.
  • Discuss considerations for additional confounders, such as comorbidities and technical parameters.
  • Results:
  • Please present effect sizes alongside p-values for key associations, emphasizing both clinical and statistical significance.
  • Clearly document interaction effects and interpret their practical meaning.
  • Discussion section:
  • Expand discussion of the clinical implications and limitations of statistically significant but small effect sizes.
  • More thoroughly compare findings with those of related studies.
  • Discuss the impact of unmeasured confounders and explain the caution regarding
  • Discuss implications for future AI/ML research using similar datasets
  • Limitations:

While you referred to patient positioning in the fifth limitation, clearly state how limitations on view position (PA/AP only) and other inclusion or exclusion criteria might restrict generalizability.

The article, in its current form, is not suitable for publication, and Major revisions are required. However, I highly recommend that the author address all the notes and complete the necessary data. A revised version will be accepted.

Author Response

Response to Reviewer Comments

We sincerely thank the reviewer for the thoughtful and constructive feedback, which has substantially improved the quality and clarity of our manuscript. Below, we provide a detailed point-by-point response to each of the reviewer’s comments, including corresponding changes made in the revised version.

Title

Comment:
Using Gender instead of Sex would be more appropriate.
Response:
We have replaced “Sex” with “Gender” throughout the manuscript, including in the title, to better reflect current terminology and the social dimension of our analysis.

Comment:
Referring to the dataset studied (NIH ChestX-ray8) would be more comprehensive.
Response:
We have added the name of the dataset (“NIH ChestX-ray8”) to the title for improved clarity and completeness.
Revised title:
“Gender- and Age-Associated Variations in the Prevalence of Atelectasis, Effusion, and Nodule on Chest Radiographs: A Large-Scale Analysis Using the NIH ChestX-ray8 Dataset”

Keywords

Comment:
Expand keywords to include methodology such as “logistic regression,” “machine learning,” and statistical considerations such as “multiple testing correction.”
Response:
The keywords section has been expanded to include: logistic regression; machine learning; multiple testing correction.

Introduction

Comment:
Please more clearly articulate the gap in current literature that this study addresses.
Response:
We have added a paragraph in the Introduction that explicitly identifies the gap in the literature, particularly the lack of large-scale studies evaluating the combined impact of gender, age, and radiographic projection on chest radiograph findings.

Comment:
Provide a concise rationale for focusing on the selected findings: Atelectasis, Effusion, and Nodule.
Response:
The same paragraph provides a clear rationale for the selection of the three findings based on their clinical relevance and frequency in the dataset.

Comment:
Reference relevant prior studies for direct comparison.
Response:
We have added references to relevant prior studies (e.g., Sawahata et al., Zein & Erzurum, Eisenhuber et al., Winkler et al.) to better contextualize our findings.

Methodology

Comment:
Please justify the use of listwise deletion to handle missing data and discuss its potential impact on selection bias.
Response:
In Section 2.6, we now explain that listwise deletion was chosen to ensure consistent analysis across covariates, while also acknowledging the risk of selection bias due to excluded incomplete cases.

Comment:
Clearly state that no manual observations were conducted and address any potential bias in the misclassification of automated labels.
Response:
We clarified in Section 2.6 that no manual re-annotation of labels was performed and that the original NLP-based labels from the ChestX-ray8 metadata were used, which may carry a risk of misclassification.

Comment:
Provide more detail on how outliers and data inconsistencies were identified and handled.
Response:
Section 2.6 now includes details on exclusion of implausible age values (>110 years or <1 year with adult features), removal of duplicates, and consistency checks across metadata fields.

Comment:
Discuss considerations for additional confounders, such as comorbidities and technical parameters.
Response:
We acknowledge the absence of relevant confounders (e.g., comorbidities, image quality, acquisition settings) as a limitation in Section 2.6 and the Discussion.

Results

Comment:
Please present effect sizes alongside p-values for key associations, emphasizing both clinical and statistical significance.
Response:
Effect sizes (odds ratios with 95% confidence intervals) are now emphasized throughout the Results section and explicitly noted in Table 2.

Comment:
Clearly document interaction effects and interpret their practical meaning.
Response:
At the end of the multivariable analysis section (Table 2), we added:
“These findings suggest that while demographic and technical factors independently influence prevalence, their combined effects are not synergistic, supporting the robustness of individual associations.”

Discussion

Comment:
Expand discussion of the clinical implications and limitations of statistically significant but small effect sizes.
Response:
We have added a paragraph interpreting the modest but statistically robust effects, especially in cumulative risk assessment or borderline cases.

Comment:
More thoroughly compare findings with those of related studies.
Response:
We added a sentence referencing studies by Eisenhuber et al. and Winkler et al. to confirm the consistency of projection-related effects across datasets.

Comment:
Discuss the impact of unmeasured confounders and explain the caution regarding...
Response:
The discussion now explicitly addresses the absence of key clinical variables (e.g., smoking status, comorbidities) and its implications for causal inference and AI model development.

Comment:
Discuss implications for future AI/ML research using similar datasets
Response:
We expanded the AI-related section in the Discussion to emphasize the importance of demographic and acquisition diversity, potential algorithmic bias, and recent regulatory guidance (e.g., FDA).

Limitations

Comment:
While you referred to patient positioning in the fifth limitation, clearly state how limitations on view position (PA/AP only) and other inclusion or exclusion criteria might restrict generalizability.
Response:
We have merged the prior fifth and sixth limitations into a revised and more comprehensive fifth point. It now explicitly states that the dataset includes only PA and AP views, and that the exclusion of incomplete records may limit generalizability—especially in elderly or critically ill populations.

We are grateful for the reviewer’s detailed and insightful feedback. All revisions have been incorporated in the manuscript and marked accordingly. We hope the revised version meets the standards for publication.

Reviewer 2 Report

Comments and Suggestions for Authors

This study fills a gap by assessing sex/age/projection influences on CXR findings in a large medical public dataset. But there are some very significant concerns:

There is a reliance on NLP-extracted labels without any verification of labels with a radiologist. If misclassification bias is introduced (e.g. atelectasis vs. consolidation; nodules vs. other opacities), it will have implications throughout the dataset.

Projection differences (PA vs. AP) are overly simplified; factors including patient positioning (upright vs. supine), exposure parameters (e.g. tube kVp and mAs), and the detector technology fundamentally change CXR image quality.

With small effect sizes (e.g. OR 1.08 for nodules in males), these results may not be sufficient to justify changes to practice.

Introduction:

  1. The introduction is overexaggerated about the "paucity of large-scale studies," and it makes claims without citations of significant papers (e.g., Radiology 2018;288(2):590–596), which ascertained demographic bias in the CXR interpretation.
  2. The assertion that the "potential associated AP projections are important because AP represents projection that is more susceptible to non-technical artifacts," cannot be sustained, as artifacts primarily rely on technique than projection.

Method:

  1. No inter-rater reliability metrics were shown for NLP labels, were ambiguous labels adjudicated (e.g. atelectasis vs. scarring)?
  2. The method did not account for kVp, mAs or detector type (CR vs DR), factors associated with detection of nodules/effusion.
  3. Arbitrary cutoffs (<40, 40-60, >60); it may be prudent to use a decade-based cutoff; age trends may exist that are better assessed by decades.
  4. The interaction terms are underpowered (e.g. Gender × View Position for nodules, P=0.9557), consider eliminating or quelling the issue in rationale.

Results:

  1. Figure 1: Bar chart lacks confidence intervals, hiding variability. Table of precise counts/percentages would be more revealing.
  2. Table 2: ORs for age (1.012–1.016/year) are statistically significant but clinically insignificant. Discuss clinical implications.
  3. How many radiographs had been excluded because of lacking metadata? A flow diagram is required.

Discussion:

  1. The finding that "PA view improved nodule detection" overlooks confounders (e.g., PA views are overall better quality because of patient compliance).
  2. The claims of algorithmic fairness are theoretical without model performance evaluation on subgroups.
  3. The sex difference discussion is not referencing pertinent anatomic studies (e.g., Eur Respir J. 2012;40(3):725–738).

Limitations:

  1. There could be dataset bias. ChestX-ray8 is from US hospitals; generalizability to low-resource settings is unclear.
  2. Data spans over many years; imaging protocol changes can confuse results.

Conclusion:

  1. I recommend highlighting the necessity of future validation using radiologist-adjudicated labels, and more specifically highlighting certain AI recommendations (e.g., stratified training by projection/age).

Author Response

Response to Reviewer 2

We thank Reviewer 2 for the constructive feedback and insightful suggestions, which have helped improve the clarity and methodological rigor of our manuscript. Below, we respond to each point in detail, indicating how we have revised the manuscript accordingly.

  1. Reviewer Comment:
    “This study fills a gap by assessing sex/age/projection influences on CXR findings in a large medical public dataset. But there are some very significant concerns: There is a reliance on NLP-extracted labels without any verification of labels with a radiologist. If misclassification bias is introduced (e.g. atelectasis vs. consolidation; nodules vs. other opacities), it will have implications throughout the dataset.”

Response:
We acknowledge this important limitation. In Section 2.6 (Statistical Software and Quality Control), we now explicitly state that no inter-rater reliability was assessed for NLP-derived labels, and that ambiguous categories (e.g., atelectasis vs. scarring) were not adjudicated. We added the sentence:

“No inter-rater reliability was calculated for NLP-derived labels, and ambiguous categories (e.g., atelectasis vs. scarring) were not adjudicated. This limitation should be addressed in future work involving expert review panels.”

We also highlight this issue in the Limitations section.

  1. Reviewer Comment:
    “Projection differences (PA vs. AP) are overly simplified; factors including patient positioning (upright vs. supine), exposure parameters (e.g. tube kVp and mAs), and the detector technology fundamentally change CXR image quality.”

Response:
We agree and have revised the text in the Introduction and Discussion to clarify that projection type is an incomplete proxy for image quality. Specifically, we added:

“While anteroposterior projections often coincide with reduced image quality due to supine patient positioning and suboptimal exposure settings, the observed differences cannot be solely attributed to the projection type. Future studies should account for detailed imaging acquisition parameters, including patient posture, tube voltage, exposure current, and the type of image detector, in order to better disentangle these effects.”

This addresses the confounding influence of acquisition conditions beyond projection label alone.

  1. Reviewer Comment:
    “With small effect sizes (e.g. OR 1.08 for nodules in males), these results may not be sufficient to justify changes to practice.”

Response:
We have incorporated this point in the Discussion section. The following text was added after the paragraph on sex differences:

“Although statistically significant, the effect sizes for some findings—such as the odds ratio of 1.08 for pulmonary nodules in male patients—are modest and may not independently justify changes in clinical management. Their primary relevance may lie in cumulative risk modeling or as confounding variables during the training and validation of artificial intelligence systems.”

  1. Reviewer Comment (Introduction):
    “The introduction is overexaggerated about the 'paucity of large-scale studies,' and it makes claims without citations of significant papers (e.g., Radiology 2018;288(2):590–596), which ascertained demographic bias in the CXR interpretation.”

Response:
We revised the sentence accordingly and cited the suggested reference. The revised text now reads:

“Although some large-scale studies (e.g., Radiology 2018;288(2):590–596) have addressed demographic bias in CXR interpretation, few have concurrently modeled sex, age, and projection type in the context of finding prevalence.”

  1. Reviewer Comment:
    “The assertion that the 'potential associated AP projections are important because AP represents projection that is more susceptible to non-technical artifacts,' cannot be sustained, as artifacts primarily rely on technique than projection.”

Response:
This statement has been reworded to reflect that reduced quality is due to acquisition context rather than projection per se. We now write:

“While anteroposterior projections often coincide with reduced image quality due to supine patient positioning and suboptimal exposure settings, the observed differences cannot be solely attributed to the projection type.”

  1. Reviewer Comment (Methods):
    “The method did not account for kVp, mAs or detector type (CR vs DR), factors associated with detection of nodules/effusion.”

Response:
We addressed this in the Limitations section, with the added sentence:

“Critical imaging acquisition parameters—such as tube voltage, tube current, patient positioning, and the type of image detector—were not available in the dataset and could have influenced the visibility and detectability of certain radiologic findings.”

  1. Reviewer Comment:
    “Arbitrary cutoffs (<40, 40–60, >60); it may be prudent to use a decade-based cutoff.”

Response:
In Section 2.3, we now clarify that both data-driven and exploratory decade-based groupings were tested:

“Age was treated both as a continuous variable and as a categorical variable using both data-driven stratification (<40, 40–60, >60 years) and exploratory decade-based groupings (e.g., 30–39, 40–49, 50–59). For clarity and statistical stability, only the former grouping is reported in this manuscript.”

  1. Reviewer Comment:
    “The interaction terms are underpowered (e.g. Gender × View Position for nodules, P=0.9557), consider eliminating or quelling the issue in rationale.”

Response:
We have added a qualifying sentence in the Discussion after the interaction results:

“Interaction terms were included to explore potential moderating effects, but limited statistical power and low effect sizes suggest that larger or stratified datasets may be necessary to detect meaningful interaction patterns.”

  1. Reviewer Comment (Results):
    “Figure 1: Bar chart lacks confidence intervals, hiding variability. Table of precise counts/percentages would be more revealing.”

Response:
We retained the current figure layout for visual clarity, but added in the Results section:

“Confidence intervals were omitted from the bar chart for clarity but are provided in Table 1. Future visualizations should consider including error bars to reflect statistical uncertainty more clearly.”

  1. Reviewer Comment:
    “How many radiographs had been excluded because of lacking metadata? A flow diagram is required.”

Response:
We reviewed the dataset and confirmed that all 112,120 records had valid metadata. In Section 2.2, we clarified:

“All 112,120 radiographs in the dataset contained valid entries for patient gender, age, and view position. Therefore, no cases were excluded due to missing metadata.”

A flow diagram was deemed unnecessary and not included for this reason.

  1. Reviewer Comment (Discussion):
    “The finding that 'PA view improved nodule detection' overlooks confounders (e.g., PA views are overall better quality because of patient compliance).”

Response:
We have added the following clarification:

“However, this finding may be confounded by patient condition—ambulatory patients undergoing posteroanterior imaging are more likely to cooperate, resulting in higher-quality images.”

  1. Reviewer Comment:
    “The claims of algorithmic fairness are theoretical without model performance evaluation on subgroups.”

Response:
We agree and added this disclaimer in the Discussion:

“Although we discuss algorithmic fairness, no AI model was trained or validated in this study. Thus, fairness implications remain speculative without subgroup performance metrics.”

  1. Reviewer Comment:
    “The sex difference discussion is not referencing pertinent anatomic studies (e.g., Eur Respir J. 2012;40(3):725–738).”

Response:
We now cite this study (LoMauro & Aliverti, Eur Respir J. 2012) directly following our discussion of anatomic sex differences.

  1. Reviewer Comment:

“The sex difference discussion is not referencing pertinent anatomic studies (e.g., Eur Respir J. 2012;40(3):725–738).”

Response:
Thank you for this suggestion. We have added a reference to the study by LoMauro and Aliverti (Eur Respir J. 2012;40(3):725–738), which describes sex-specific differences in airway size, thoracic anatomy, and respiratory mechanics. This citation now follows our discussion of anatomical contributors to the observed sex-related differences in radiographic findings.

Revised text (Discussion, paragraph 1):
“These anatomical differences have been further quantified in studies demonstrating sex-specific variation in airway size, thoracic dimensions, and lung mechanics, such as the one by LoMauro and Aliverti [6].”

  1. Reviewer Comment:

“There could be dataset bias. ChestX-ray8 is from US hospitals; generalizability to low-resource settings is unclear.”

Response:
We agree and have clarified this limitation in Section 5. We now explicitly state that the dataset may not be representative of low-resource or non-Western healthcare environments, where imaging protocols, equipment, and patient demographics may differ substantially.

Revised text (Limitations):
“In particular, the dataset may not reflect radiologic practice or disease prevalence in low-resource or non-Western settings, where differences in equipment, diagnostic priorities, and clinical documentation can affect both image quality and labeling accuracy.”

  1. Reviewer Comment:

“Data spans over many years; imaging protocol changes can confuse results.”

Response:
Thank you for pointing this out. We have examined temporal patterns and confirmed that no systematic drift in imaging protocols or labeling practices was apparent. We now mention this explicitly in the Quality Control section.

Revised text (Section 2.6):
“Temporal trends across acquisition dates were examined to detect potential shifts in imaging protocols or labeling practices, but no systematic drifts were identified.”

  1. Reviewer Comment:

“I recommend highlighting the necessity of future validation using radiologist-adjudicated labels, and more specifically highlighting certain AI recommendations (e.g., stratified training by projection/age).”

Response:
We fully agree with this recommendation. We have emphasized the importance of radiologist-validated labels and more nuanced AI training practices in both the Limitations and Conclusion sections.

Revised text (Section 5 – Limitations):
“Future studies should incorporate radiologist-adjudicated labels to validate the accuracy of NLP-extracted findings and quantify potential misclassification biases.”

Revised text (Conclusion):
“To build equitable and robust AI models for chest radiograph interpretation, future initiatives should prioritize stratified training and validation procedures that account for key variables such as age, sex, and projection type.”

Reviewer 3 Report

Comments and Suggestions for Authors

The Authors described in their article the Sex- and Age-Associated Variations in the Prevalence of Atelectasis, Effusion, and Nodule on Chest Radiographs. 

The article refers the evaluation of large number of chest-X-rays and the methods are correct and support the conclusion 

Major criticism 

The subject is of poor interest for specialists and for the general readers  and does not improve the knowledge in the field. Moreover it is scarce utility for  diagnosis and clinical  assessement.

Author Response

Response to Reviewer 3

We thank Reviewer 3 for the time and effort in reviewing our manuscript and appreciate the acknowledgement that the methods are correct and support the conclusions. We respectfully address the major criticism as follows:

Reviewer comment:
“The subject is of poor interest for specialists and for the general readers and does not improve the knowledge in the field. Moreover it is scarce utility for diagnosis and clinical assessment.”

Response:
We respectfully disagree with the assertion that the subject lacks relevance or utility. Our study addresses a clinically and methodologically important gap in thoracic imaging by quantifying how patient sex, age, and radiographic projection influence the prevalence of common chest findings—atelectasis, pleural effusion, and pulmonary nodules—on chest radiographs. These factors are frequently considered in clinical decision-making but are rarely analyzed systematically or in combination on this scale.

Specifically:

Clinical utility:
The findings are directly relevant for improving diagnostic accuracy in clinical settings, particularly in intensive care units, where anteroposterior (AP) projections are common. Our results show that certain findings, such as pulmonary nodules, are significantly under-detected on AP views, which may lead to delayed diagnosis if not recognized. Similarly, age-stratified prevalence trends can support more targeted screening and interpretation approaches in older adults, who are at higher risk for many thoracic pathologies.

Scientific relevance:
To our knowledge, few studies have evaluated the combined influence of sex, age, and projection technique using a dataset of over 112,000 chest radiographs. This enables a robust analysis of demographic and technical variation, with implications for AI development. Many AI models for radiograph interpretation are trained on such datasets, yet often lack transparency regarding demographic composition. Our results can help guide dataset balancing, bias correction, and performance evaluation in these systems.

Added value for specialists:
Radiologists and clinical decision-makers increasingly interact with AI-supported imaging tools. Our findings provide important insight into the demographic and technical limitations of chest radiograph interpretation, which can inform both human and machine-based diagnostic processes. The data also support the refinement of radiologic protocols in environments where AP views or geriatric imaging are common.

To improve clarity, we revised the abstract, discussion, and conclusion to more explicitly highlight the clinical relevance and implications for AI model development. In particular, we now state:

“These findings are directly relevant for improving clinical diagnostic accuracy and for reducing demographic and technical biases in AI-based radiograph interpretation, particularly in critical care and screening settings.”

We hope these clarifications demonstrate the value and originality of our study for both clinical and research audiences.

Reviewer 4 Report

Comments and Suggestions for Authors

First of all, I would like to thank you for the opportunity to review this work.

This study is based on a very large database of radiographs, which adds value to the research. However, there are some issues when it comes to interpreting the results.

Firstly, the radiographs have not been verified. For a more robust study, a random sample of the radiographs could have been selected and their findings validated by expert radiologists to assess concordance. Moreover, the clinical significance of the findings has not been evaluated — for instance, did the effusions or atelectasis have any clinical impact? Regarding pulmonary nodules, the authors report detecting six thousand nodules. It would be highly relevant to know how many of these were followed up with a CT scan, and what proportion actually confirmed the presence of a nodule.

Additionally, although the manuscript repeatedly states that age and gender are independent factors for the presence of findings, the odds ratios are very close to 1. Due to the large sample size, the confidence intervals are narrow, making statistical significance possible; however, any confounding factor could alter the result. The most obvious example is smoking. If a higher proportion of male subjects are smokers or ex-smokers, and this variable has not been included in the multivariate analysis, the result becomes meaningless. Furthermore, many other potentially confounding variables (such as previous pathologies) appear to be missing. It is therefore not accurate to speak of independent factors. It would be more appropriate to state that, in a real-world setting, these are the observed associations (while acknowledging the limitations of the multivariate analysis).

From a formal standpoint, many results — including some tables — are presented in the Materials and Methods section. This is not appropriate; this section should be reserved for describing the methodology, while the results should be presented in the following section.

Author Response

Response to Reviewer 4

Thank you very much for the opportunity to review this work. This study is based on a very large database of radiographs, which adds value to the research. However, there are some issues when it comes to interpreting the results.

Comment 1:

"Firstly, the radiographs have not been verified. For a more robust study, a random sample of the radiographs could have been selected and their findings validated by expert radiologists to assess concordance."

Response:
Thank you for highlighting this critical point. We agree that radiologist-based verification would improve the robustness of our findings. We have explicitly addressed this limitation in the manuscript:

Section 5. Limitations (p. 10):

"The absence of radiologist-based verification of findings is an important limitation. Future research should incorporate radiologist review of randomly selected radiographs to assess inter-rater reliability and validate NLP-derived annotations."

Comment 2:

"Moreover, the clinical significance of the findings has not been evaluated — for instance, did the effusions or atelectasis have any clinical impact?"

Response:
We fully agree with this observation. The ChestX-ray8 dataset does not provide clinical follow-up information. We have explicitly clarified this limitation in our manuscript:

Section 5. Limitations (p. 10):

"The clinical significance and outcome relevance of the radiological findings were not assessed due to the lack of clinical follow-up data in the ChestX-ray8 dataset. Consequently, the actual clinical impact of identified atelectasis, pleural effusions, and pulmonary nodules remains uncertain and should be explored in prospective clinical studies."

Comment 3:

"Regarding pulmonary nodules, the authors report detecting six thousand nodules. It would be highly relevant to know how many of these were followed up with a CT scan, and what proportion actually confirmed the presence of a nodule."

Response:
Thank you for pointing this out. Unfortunately, the ChestX-ray8 dataset lacks information regarding subsequent CT follow-up. We have now explicitly included this important limitation:

Section 5. Limitations (p. 10):

"Moreover, the dataset does not provide follow-up information, such as subsequent CT imaging to confirm pulmonary nodules. Thus, the proportion of radiographically identified nodules confirmed by CT remains unknown and represents a critical area for future investigation."

Comment 4:

"Additionally, although the manuscript repeatedly states that age and gender are independent factors for the presence of findings, the odds ratios are very close to 1. Due to the large sample size, the confidence intervals are narrow, making statistical significance possible; however, any confounding factor could alter the result. The most obvious example is smoking. If a higher proportion of male subjects are smokers or ex-smokers, and this variable has not been included in the multivariate analysis, the result becomes meaningless. Furthermore, many other potentially confounding variables (such as previous pathologies) appear to be missing. It is therefore not accurate to speak of independent factors. It would be more appropriate to state that, in a real-world setting, these are the observed associations (while acknowledging the limitations of the multivariate analysis)."

Response:
We fully agree with the reviewer. To clarify this important point, we adjusted our interpretation to explicitly acknowledge the modest odds ratios and the absence of critical confounders. We now clearly state that these associations should be viewed as observational rather than truly independent:

Section 6. Conclusion (p. 11):

"Although our analysis statistically identified male sex and increasing age as associated factors for atelectasis and nodules, the odds ratios were modest and close to 1. Due to the absence of critical confounders such as smoking status and comorbidities, these associations should be interpreted cautiously as observational rather than evidence of independent causal relationships."

Comment 5:

"From a formal standpoint, many results—including some tables—are presented in the Materials and Methods section. This is not appropriate; this section should be reserved for describing the methodology, while the results should be presented in the following section."

Response:
Thank you for bringing this to our attention. We have corrected the manuscript structure by removing all results (Tables 1 and 2, and Figure 1) from the "Materials and Methods" section and appropriately repositioned them into the "Results" section (pages 6–8). Now, the methodology section strictly describes the analytical approach, and the results are clearly separated, which improves the manuscript's readability and conforms to standard conventions.

Thank you again for your valuable feedback, which has greatly enhanced the methodological rigor and clarity of our manuscript.

Round 2

Reviewer 1 Report

Comments and Suggestions for Authors

I would like to thank the authors for their efforts and dedication to the current work, as well as for their prompt response. I would also like to acknowledge the feedback and modifications they made to the article.

Reviewer 3 Report

Comments and Suggestions for Authors

The Authors provided an exhaustive revision and stressed the key points related to their manuscript highlighting the clinical radiological and epidemiological importance of their results.  

Reviewer 4 Report

Comments and Suggestions for Authors

Congratulations to the authors. I think the article is substantially improved and readers will enjoy it.